# Gut and Endometrial Microbiome Dysbiosis: A New Emergent Risk Factor for Endometrial Cancer

**DOI:** 10.3390/jpm11070659

**Published:** 2021-07-14

**Authors:** Soukaina Boutriq, Alicia González-González, Isaac Plaza-Andrades, Aurora Laborda-Illanes, Lidia Sánchez-Alcoholado, Jesús Peralta-Linero, María Emilia Domínguez-Recio, María José Bermejo-Pérez, Rocío Lavado-Valenzuela, Emilio Alba, María Isabel Queipo-Ortuño

**Affiliations:** 1Unidad de Gestión Clínica Intercentros de Oncología Médica, Hospitales Universitarios Regional y Virgen de la Victoria, Instituto de Investigación Biomédica de Málaga (IBIMA)-CIMES-UMA, 29010 Málaga, Spain; soukaina@ibima.eu (S.B.); alicia.gonzalez@ibima.eu (A.G.-G.); isaac.plaza.andrades@ibima.eu (I.P.-A.); aurora.laborda@ibima.eu (A.L.-I.); l.sanchez.alcoholado@ibima.eu (L.S.-A.); jesus.peralta@ibima.eu (J.P.-L.); emilia.dominguez@ibima.eu (M.E.D.-R.); cheberpe@gmail.com (M.J.B.-P.); maribel.queipo@ibima.eu (M.I.Q.-O.); 2Instituto de Investigación Biomédica de Málaga (IBIMA), Campus de Teatinos s/n, 29071 Málaga, Spain; 3Facultad de Medicina, Universidad de Málaga, 29071 Málaga, Spain; 4Centro de Investigación Biomédica en Red de Cáncer (Ciberonc CB16/12/00481), 28029 Madrid, Spain

**Keywords:** endometrial cancer, endometrial microbiome, gut microbiome, dysbiosis, estrogen metabolism, estrobolome, inflammation, antitumour treatment, prebiotics, probiotics

## Abstract

Endometrial cancer is one of the most common gynaecological malignancies worldwide. Histologically, two types of endometrial cancer with morphological and molecular differences and also therapeutic implications have been identified. Type I endometrial cancer has an endometrioid morphology and is estrogen-dependent, while Type II appears with non-endometrioid differentiation and follows an estrogen-unrelated pathway. Understanding the molecular biology and genetics of endometrial cancer is crucial for its prognosis and the development of novel therapies for its treatment. However, until now, scant attention has been paid to environmental components like the microbiome. Recently, due to emerging evidence that the uterus is not a sterile cavity, some studies have begun to investigate the composition of the endometrial microbiome and its role in endometrial cancer. In this review, we summarize the current state of this line of investigation, focusing on the relationship between gut and endometrial microbiome and inflammation, estrogen metabolism, and different endometrial cancer therapies.

## 1. Introduction

The endometrium is a very dynamic tissue that undergoes proliferation and differentiation processes during the menstrual cycle in response to variations in the levels of steroid sex hormones (estrogen and progesterone) produced in the ovaries, and the release of local factors [1].

Endometrial cancer is the sixth most common malignancy in women, and the fifteenth most common cancer [2]. It accounts for nearly 5% of total cancer cases and more than 2% of cancer deaths among women worldwide [3]. In the United States and some European countries, the incidence of endometrial cancer is higher than in other developed countries, being the fourth most common cancer in women, accounting for approximately 6% of new cancer cases and 3% of cancer deaths each year [4].

This high incidence in the United States and Europe compared to other countries may be due to high rates of obesity, as well as other important risk factors such as advanced age, early menarche, late menopause, nulliparity, and post-menopausal estrogen therapy [5]. Endometrial cancer occurs more frequently after menopause and is generally associated with a good prognosis [6].

Whereas high parity, late age at last birth, physical activity, the use of combined oral contraceptives and tabacco consumption are considered factors with a protective role against endometrial cancer [7], there are other several factors that increase the risk of endometrial cancer, such as obesity, the use of hormone replacement therapy (HRT) to treat menopausal symptoms, and a family history of cancers such as Lynch syndrome (an autosomal dominant disorder characterized by juvenile onset of malignant tumours and colorectal cancer). Women with Lynch syndrome have an increased endometrial cancer risk as well as an increased risk for other types of cancer such us colorectal cancer [8]. This syndrome is caused by a loss-of-function germline mutation in one of four genes (human mutL homolog 1 (*MLH1*), *MSH2*, *MSH6*, and *PMS1* Homolog 2 (*PMS2*)) involved in mismatch-pair recognition and initiation of repair [9]. *MLH1* and *MSH2* mutations are more frequent (60–80%) in patients with lynch syndrome comparated to *MSH6* and *PMS2* mutations. Mutation in epithelial cellular adhesion molecule (*EPCAM*) (gene located in *MSH2* gene promoter and that lead to its epigenetic inactivation) is also identificated in lynch syndrome. The mismatch repair genes inactivation induces accumulation of different gene mutations, leading to cancer development with microsatellite instability phenotype [10].

Bokhman was the first to classify endometrial cancer into two different histological types in 1983 [11]. This classification, into Type I and Type II endometrial cancer, has revealed the existence of differences in molecular characteristics, which consequently translate into differences in prognosis and treatment [12]. In Bokhman’s study, the frequency of the first pathological type (Type I) in the group of women studied was 65%, while the frequency of the second type (Type II) was 35% [13].

Endometrioid, or Type I endometrial cancer generally originates in a hyperplastic endometrial context [14], expressing estrogen and progesterone receptors, and is therefore typically associated with hormonal disorders [15]. In addition to phosphatase and tensin homolog (*PTEN*) and phosphatidylinositol 3-kinase (*PIK3CA*) mutations, which are the most common in Type I endometrial cancer, other mutations have been identified in *KRAS* and cadherin associated protein (*β-catenin*) genes [16]. In some Type I endometrial cancers, mutations that inactivate *MSH6* have been identified as being associated with microsatellite instability [17] (Figure 1).

Non-endometrioid, or Type II endometrial cancer, is less common, accounting for approximately 10–20% of endometrial cancer cases [18]. Type II endometrial cancer develops in an atrophic endometrial context, histologically poorly differentiated, with a tendency towards a deep invasion into the myometrium, and a high frequency of metastasis [19]. Type II endometrial cancer is characterized by a high number of tumour suppressor *p53* mutations [20], and other low-frequency genomic alterations such as tumour suppressor cyclin-dependent kinase inhibitor 2A (*p16*) inactivation and Erb-B2 receptor tyrosine kinase 2 (*HER-2/neu*) over-activation [17]. Type II endometrial carcinoma includes carcinosarcomas, serous and clear cell carcinomas, and mixed Mullerian tumours [21] (Figure 1).

However, genetic alterations alone are not enough to explain the origins of endometrial cancer; other environmental factors such as hormones, obesity, and diabetes also have an influence, as does the microbiome, which comprises an important part of the uterine microenvironment [22]. Nevertheless, the molecular mechanisms involved in the interaction between microbiome and endometrial cancer still need further elucidation.

## 2. Endometrial Microbiome

Previously, it had long been thought that the human uterus was a sterile environment free of microorganisms. However, recent studies using molecular techniques have confirmed the existence of microbiota in the endometrium, playing an important role in the proper functioning of the endometrium and in the development of pregnancy under normal conditions [23].

In the vagina, the microbiota has an important preventive role against various urogenital diseases, such as bacterial vaginosis, fungal infections, sexually transmitted infections, urinary tract infections, and HIV. This protective role is mainly due to the production of lactic acid by *Lactobacillus* species (spp.), which are commonly associated with a healthy vagina, and which produce several bacteriostatic and bactericidal components that help to lower the pH of the vaginal microenvironment and promote competitive exclusion [24]. However, the composition of vaginal microbiota varies between the different phases of menopause (pre-, peri-, and post-menopause) and also in pathological conditions such as vaginal atrophy in which the abundance of *Lactobacillus* decreases while *Anaerococcus*, *Peptoniphilus*, and *Prevotella* levels increase [25].

Until now, few studies have been devoted to investigating the composition of endometrial microbiota, so it is a subject that remains quite obscure today.

Many of the studies done have confirmed that, as in the vagina, *Lactobacillus* is the dominant genus in the endometrium. In fact, Mitchell et al. compared vaginal microbiota with uterine microbiota and found that the endometrium is characterized by the presence of *Lactobacillus*, being the most abundant genus, followed by *Gardnerella*, *Prevotella*, *Atopobium,* and *Sneathia* [26]. Fang et al. compared the bacterial composition of the vagina with that of the endometrium, and also compared endometrial microbiome composition between patients in different situations including healthy women, patients with endometrial polyps, and patients with chronic endometritis. They found that Proteobacteria, Firmicutes, and Actinobacteria dominated the intrauterine microbiome in all the studied groups. Furthermore, although they found significant differences between the vaginal and the endometrial microbiome, at the genus level, *Lactobacillus*, *Gardnerella*, *Bifidobacterium*, *Streptococcus*, and *Alteromonas* were significantly higher in the healthy group when compared with the others [27].

However, in contrast to these studies, others have suggested that non-*Lactobacillus* species are more common in the endometrium. In 2016, Verstraelen et al. found that 90% of the women included in their study had an endometrial microbiota profile dominated by *Bacteroides* (*Bacteroides xylanisolvens*, *Bacteroides thetaiotaomicron*, and *Bacteroides fragilis*) and *Pelomonas* [28]. Chen et al. confirmed the existence of different bacterial communities throughout the female reproductive system, with a continuous change from the vagina to the ovaries, with *Pseudomonas*, *Acinetobacter*, *Vagococcus*, and *Sphingobium* being the most abundant in the endometrium [29]. Winters et al. also sequenced endometrial samples from 25 women who underwent total hysterectomy for fibroids or endometrial hyperplasia, and found that the most abundant genera in their endometria were *Acinetobacter*, *Pseudomonas*, *Comamonadaceae*, and *Cloacibacterium* [30]. Finally, in a more recent study, Lu et al. also suggested that *Lactobacillus* is not the predominant genus in the endometrium, observing a greater abundance of *Rhodococcus*, *Phyllobacterium*, *Sphingomonas*, *Bacteroides*, and *Bifidobacterium* [31] (Table 1).

## 3. Microbiome and Endometrial Cancer

Currently, there is abundant evidence demonstrating the involvement of bacteria in the development and expansion of various pathologies, including different types of cancers [32]. Many of the bacteria that colonize the human body establish beneficial relationships with their host. Notwithstanding, dysbiosis promotes the development of various diseases [33].

For example, in the stomach, *Helicobacter pylori* is one of the most common human infectious agents that produces several virulence factors linked to significant disorders in the host’s intracellular signalling pathways, favouring the appearance of neoplastic transformations. Consequently, infection by this bacterium is considered to be an important risk factor for gastrointestinal cancer [34]. Furthermore, in cervical (cervix) cancer, human papillomavirus is a known cause of the disease [35].

It is known that the pathogenesis of endometrial cancer involves mainly an excess of estrogen levels, and there is also evidence that the composition of the microbiota may be an important risk factor, given the inflammatory profile in endometrial cancer. However, the composition of endometrial microbiota in endometrial cancer remains poorly studied.

Accordingly, Walther-António et al. identified the differences in the composition of endometrial microbiota in different diseases. In women with abnormal bleeding and endometritis they found *Shigella* and *Barnesiella* to be the most dominant genera in their endometria. They also observed that there is a difference in the composition of the endometrial microbiota under normal conditions as compared with hyperplasia, which suggests a role for microbiota in the early phases of cell transformation, since hyperplasia is considered to be a precancerous transformation of the endometrium. To test this hypothesis, they compared the microbiota of patients with endometrial hyperplasia and those with endometrial cancer, and found no significant differences. Furthermore, as a result of sequencing samples from endometrial cancer patients, they saw that the taxa Firmicutes (*Anaerostipes*, *ph2*, *Dialister*, *Peptoniphilus*, *1–68*, *Ruminococcus*, and *Anaerotruncus*), Spirochaetes (*Treponema*), Actinobacteria (*Atopobium*), Bacteroidetes (*Bacteroides* and *Porphyromonas*), and Proteobacteria (*Arthrospira*) were enriched. However, the most prevalent result they obtained is the close correlation between the species *Atopobium vaginae* and *Porphyromonas sp*. and endometrial cancer, especially when the vaginal pH is high (˃4.5) [22]. *Porphyromonas gingivalis* is considered to be a biomarker of death risk from aerodigestive cancer, independent of periodontal diseases [36]. Based on the relationship between this bacteria and various pathologies, Walther-António et al. predicted the possible involvement of *Porphyromonas* spp. in the progression of the processes leading to the development of endometrial cancer. Furthermore, knowing that *Atopobium vaginae* is associated with bacterial vaginosis, they hypothesized that it may be involved in creating chronic inflammation that leads to local immune dysregulation, thus facilitating intracellular infection by *Porphyromonas* species [22]. Moreover, *Porphyromonas* spp. combined with high pH in the vagina could be a promising biomarker for endometrial cancer [37].

Lu et al. demonstrated that there is a difference in the composition of endometrial microbiota between patients with endometrial cancer and patients with benign uterine lesions. In the results they obtained, a decrease in the local diversity of the microbiota was observed in the group of patients with endometrial cancer compared to patients with benign uterine lesions. This decrease in the diversity of microorganisms led to the overgrowth of the few remaining species and a decrease in resilience. Furthermore, in this study, *Pseudomoramibacter_Eubacterium*, *Rhodobacter*, *Vogesella*, *Bilophila*, *Rheinheimera*, and *Megamonas* were enriched in patients with benign uterine lesions, while *Micrococcus* was associated with an inflammatory profile in endometrial cancer [31].

With the aim of studying the possible differences in bacterial, archaea, and viral transcript (BAVT) in different gynaecological cancers and in normal fallopian tubes, Gonzalez-Bosquet et al. carried out a metagenomic analysis of high-grade serous carcinoma (HGSC) and endometrioid endometrial carcinoma (EEC), and compared them with normal fallopian tubes. They found that there were 93 BAVTs differentially expressed between HGSC and EEC. However, 12 BAVT species were independently expressed in all the samples, and 6 of them were also significantly expressed (*Pusillimonas* sp. *ye^3^*, *Riemerella anatipestifer*, *Salinibacter ruber*, *Bacillus tropicus*, *Nostocales cyanobacterium HT-58-2*, and *Corynebacterium pseudotuberculosis*). Nevertheless, in normal samples these BAVT species were the highest, while they were decreased in EEC, and even more so in HGSC samples.

Gonzalez-Bosquet et al. also investigated the origins of these BAVTs, and they saw some human loci that harbour genetic material from these microorganisms; more exactly BAVTs were located within or close to genes or lncRNAS [38].

Walsh et al., like Walther et al., identified *Porphyromonas somerae* as the most abundant organism in patients with endometrial cancer. They also found that in addition to obesity and postmenopause, a high vaginal pH is considered an additional risk factor for endometrial cancer. In that study, they confirmed that *Porphyromonas somerae* is not associated with postmenopause; it is, however, related to four other microorganisms (*Anaerococcus tetradius*, *Anaerococcus lactolyticus*, *Peptoniphilus coxii*, and *Campylobacter ureolyticus*) which are associated with postmenopause, suggesting that they could be first-colonizers to facilitate the subsequent colonization by *Porphyromonas somerae* and others. *Porphyromonas somerae* was found in 100% of samples from patients with Type II endometrial cancer and in 57% of patients with endometrial hyperplasia. Therefore, *Porphyromonas somerae* is considered to be a biomarker of the disease.

In contrast to the results obtained by Lu et al. which affirm that the diversity of the local microbiota decreases in endometrial cancer, Walsh et al. observed that the risk factors for endometrial cancer (postmenopause, obesity, and high vaginal pH) increase the diversity of endometrial microbiota. Walsh et al. identified seventeen enriched taxa in patients with endometrial cancer, eight of which were enriched by menopause. Because of the prominence of postmenopause as a risk factor for endometrial cancer, it could be considered as one of the main conditions that favour the disease state [39]. In postmenopausal women, the production of ovarian estrogens ceases, leading to a decrease in glycogen levels, which induces a decrease in colonization by *Lactobacillus*, basifying the pH of the medium. Under normal conditions, *Lactobacillus* produces lactic acid that contributes to the maintenance of the low pH of the vagina, so it can act as a selective barrier to the rest of the reproductive system, avoiding its colonization by pathogens and helping to maintain the microbiota specific to each part of the reproductive system [40].

According to circumstances, species favoured by menopause are not directly involved in endometrial cancer, but they are probably facilitating colonization by other species associated with endometrial cancer [39]. *Atopobium vaginae* is a characteristic pathogen of bacterial vaginosis [41]. It is conceivable that some women with endometrial cancer may have been previously diagnosed with bacterial vaginosis, which could explain the association of this bacteria with endometrial cancer [39] (Table 2).

## 4. Estrogen Metabolism, Gut Microbiota, and Endometrial Cancer

Estrogens play a major role in modulating the growth of the endometrium by inducing proliferation at the end of the proliferative phase of the menstrual cycle [42]. However, after ovulation, in the luteal phase, the increase in estrogen levels induces the production of progesterone which inhibits the proliferation of the endometrium, and promotes its transition to a receptive state, preparing it for the implantation of a blastocyst. An increase in estrogen levels leads to an imbalance between the production of estrogens and progesterone, favouring the appearance and development of endometrial cancer [43].

Endometrioid carcinoma accounts for 80% of all endometrial cancer cases. This type of endometrial cancer is fundamentally caused by excessive exposure to estrogens which, in the absence of the counteractive effects of progesterone, induces endometrial proliferation and therefore endometrial hyperplasia and ultimately cancer [44].

Estrogens are produced in the ovaries and are then transported to the organs, including the uterus and breasts, where they have different functions. Later, they are transported to the liver where they are metabolized to facilitate their elimination from the body. The estrogenic hormones, estrone (E1) and estradiol (E2), undergo irreversible hydroxylation at the C-2, C-4, or C-16 carbon of the steroid ring. Estrogen metabolites are conjugated by sulfonation or glucuronidation, producing changes in their structure and bioavailability. Conjugated estrogens are excreted in urine or bile. Finally, the inactive conjugated estrogens excreted in the bile are transported to the distal part of the intestine to be eliminated through faeces [45].

However, inactive estrogens in the intestine are occasionally activated by deconjugation and are reabsorbed through the intestinal mucosa and enter the bloodstream via the portal vein (Figure 2). It has been established that the intestinal microbiota is involved in the reactivation of estrogens and, therefore, in the regulation of estrogen levels [46]. Estrobolome was defined for the first time in 2011 as an aggregate of genes from enteric bacteria, whose products are capable of metabolizing estrogens, specifically, bacteria with β-glucuronidase activity, an enzyme involved in the deconjugation of estrogens [47].

β-glucuronidase activity plays a significant role in the generation of toxic and carcinogenic metabolites in the intestine, and also in the reabsorption of various compounds into the circulatory system, such as estrogens [48]. β-glucuronidase facilitates binding to estrogen receptors, and the activation of these receptors increases the number of cells in the G0/G1 phase of the cell cycle, promoting proliferation, a process well described in breast cancer, highlighting the relationship existing between gut microbiota and estrogen levels in breast cancer.

Like breast cancer, endometrial cancer is also considered to be hormone-dependent, in which the intestinal microbiota is involved, especially in obese patients. There is a possibility that the composition and diversity of the microbiota favours bacteria capable of metabolizing estrogens, which allows greater reabsorption of estrogens and increased binding to receptors, contributing to the development of endometrial hyperplasia and endometrial cancer [49].

An active estrobolome is capable of modulating endogenous estrogen metabolism by β-glucuronidase and β-glucosidase enzymatic activity, thereby controlling circulating and excreted estrogen levels. In the gastrointestinal tract, the most important genes encoding β-glucuronidase enzyme activity are the β-glucuronidase (*GUS*) genes (Figure 2). Recently, an atlas for the characterization of the β-glucuronidase of the human intestinal microbiota was compiled. Approximately 112 new *GUS* genes were identified and grouped into six classes expressed in four bacterial phyla, denominated as Bacteroidetes, Firmicutes, Verrucomicrobia, and Proteobacteria. Within them, the phylum Bacteroidetes presents a greater abundance and diversity of *GUS* enzymes [50]. β-glucuronidase activity is modulated by diet and the microbiota. A diet rich in fat or protein has been associated with high faecal levels of β-glucuronidase, while a fibre-based diet decreases the activity of this enzyme. Furthermore, the β-glucuronidase activity of *Escherichia coli* cultures is controlled by population density, suggesting the involvement of quorum sensing in the control of enzyme activity. Although it remains to be determined how β-glucuronidase and β-glucosidase contribute to breast and endometrial cancer, there is ample evidence which suggests that both play an important role [51].

In one of their studies, Flores et al. found that in postmenopausal women and in men, non-ovarian estrogen levels were closely associated with the amount and diversity of faecal microbiota, with the taxa Clostridia within Firmicutes being the most related, in addition to three genera of the Ruminococcaceae family. Furthermore, the activity of β-glucuronidase has been associated with the levels of estrone, but not estrogen, in urine, whereas in pre-menopausal women, estrogen levels are not influenced by the microbiota or by β-glucuronidase activity [46].

Therefore, estrobolome can modulate circulating estrogen levels, which can alter vaginal microbial communities. In accordance with this concept, previous studies have established that gut microbiome is indirectly involved in endometrial carcinogenesis through its altering of genital microbial communities [52]. Thus, more attention is being paid to the study of gut microbiome modulation methods to treat estrogen-dependent diseases, including endometrial cancer, through bariatric surgery, faecal bacteria transfer, the use of pharmaceutical (Metformin) and nutraceutical (Genistein) methods, which in several studies have shown favourable results in treating the metabolic aspects of the disease [49].

## 5. Microbiome, Inflammation, and Endometrial Cancer

In 1863, Rudolf Virchow discovered the existence of leukocytes in neoplastic tissues, and established a correlation between inflammation and cancer. Since then, many studies have relied on Virchow’s hypothesis to investigate cancer prevention and treatment methods [53]. There are many examples that support this hypothesis, such as hepatitis and liver cancer, or colitis and colorectal cancer. In addition, the use of non-steroidal anti-inflammatory drugs has been shown to reduce the risk of various cancers.

Even though the mechanisms by which local inflammation facilitates cancer development are unknown, the production of cytokines, such as tumour necrosis factor α (TNF-α) by local tissue and infiltrated inflammatory cells, seems to play a key role [54]. Chronic inflammation promotes angiogenesis, cell proliferation, and the production of free radicals that cause DNA damage and facilitate tumour initiation and development [55].

After menopause, when estrogen production comes to an end, most of the circulating estrogens are produced in adipose tissue through the conversion of androgens by the enzyme Aromatase [56]. IL-6 has been shown to stimulate aromatase activity in adipose tissue and in endometrial cancer stromal cells, which then increases estrogen levels [57]. Patients with endometrial cancer have high levels of IL-6 and nuclear factor-kB (NF-kB), a cellular transcription factor that activates several genes involved in the inflammatory and immune response [55]. The activation of NF-kB leads to the expression of COX-2, which induces the production of prostaglandin E2, a protein which is able to transform endometrial cells into neoplastic tissue [58].

The microbiome could be involved in the initiation of inflammation (Figure 3), inducing the immunopathological changes which ultimately lead to the development of cancer [59]. The activation of immune receptors induces the cellular response, by activating the mitogen-activated protein kinase (MAPK), NF-κB, or PI3K/AKT signalling pathways. Activation of these signalling pathways induces the expression of pro-inflammatory cytokines (e.g., TNF-α, IL-6, and IL-8) and/or antimicrobial peptides, which are involved in the development of the inflammatory response [31].

As previously mentioned, the composition of the uterine microbiome is linked to several gynaecological pathologies, such as endometriosis, dysfunctional menstrual bleeding, and cancer [60] (Figure 3). Endometrial cancer is characterized by the simultaneous presence of two species, *Atopobium vaginae* and *Porphyromonas* [22]. In 2019, Caselli et al. performed an in-vitro analysis of the effect of *Atopobium vaginae* and *Porphyromonas somerae* on the expression of pro-inflammatory cytokines in endometrial cells using HEC-1A cells (human endometrial adenocarcinoma cells). The results of this study demonstrated that 24 h was sufficient to induce production of pro-inflammatory cytokines in endometrial cells cultured with *Atopobium vaginae* and *Porphyromonas somerae*. These cytokines produced in cells cultured with *Atopobium vaginae* and *Porphyromonas somerae* are not the same as those produced in cells cultured with *Lactobacillus vaginalis*. Thus, while *Lactobacillus vaginalis* induces the production of IL-8, *Atopobium vaginae* and *Porphyromonas somerae* induce the production of IL-1α, IL-1β, IL-17α, and TNFα, but not IL-8. This cytokine production was maintained over time without significant changes, suggesting a specific kinetic of cytokine induction, or a very gradual decrease in the ability of these bacteria to induce cytokine production. In the control group used with dead bacteria, it was observed that there was no production of cytokines, highlighting the need for the presence of *Atopobium vaginae* and *Porphyromonas somerae* to stimulate cytokine production in endometrial cells [61]. Several studies have demonstrated that IL1α and IL1β are overexpressed in various tumours including endometrial cancer, and promote cell proliferation, adhesion, invasion, and angiogenesis [62]. IL-17α, in turn, induces the production of other inflammatory proteins such as IL-8 and TNFα, stimulating the proliferation of endometrial cells, contributing to endometriosis and angiogenesis [63]. TNFα has been implicated in endometrial hyperplasia and, therefore, in endometrial cancer, evidencing an important role in metastasis and resistance to chemotherapy [64]. In addition to stimulating the production of the previously mentioned cytokines, Caselli et al. found that *Atopobium vaginae* and *Porphyromonas somerae* alter the transcription of other proteins, including CCL13, CCL8, CXCL2, IL22, and IL9. However, CCL13 production was also stimulated in the presence of *Lactobacillus vaginalis*. This implies the absence of a specific relationship between the production of CCL13 and the presence of *Atopobium vaginae* and *Porphyromonas somerae*. CCL8 and CXCL2 promote tumour invasion in various types of cancers, including endometrial cancer. Chemokine CXCL2 expression is induced by TNFα [61]. Interleukin IL-9 plays an important role in the immune response and cancer pathogenesis [65]. IL-22 induces breast cancer progression and endometrial cell proliferation through the production of CCL2 and IL-8 [61].

In another study, Lu et al. found that IL-6 and IL-17 mRNA levels are positively correlated with a relative abundance of *Micrococcus*, another Gram-positive bacterium belonging to the Phylum *Actinobacteria* related to endometrial cancer [31].

## 6. Modulation of Antitumoural Therapies Efficacy and Toxicity by Gut Microbiota

Endometrial cancer is mainly treated with surgery, to determine the stage of the tumour as an initial step to identify patients who could benefit from chemotherapy or radiation therapy. Immunotherapy is increasingly being investigated and seems to have favourable results in patients with microsatellite instability (MSI). In addition, recent studies have been conducted to establish whether the inhibition of immune checkpoints could be considered as a possible antitumour treatment method [66].

However, these treatments, particularly chemotherapy and radiotherapy, are very aggressive and can cause various side effects, especially at the intestinal level. Thus, up to 80% of patients exhibit intestinal symptoms such as abdominal pain and diarrhoea, among others, during treatment [67].

Consequently, recent studies have investigated the possibility of exploiting the microbiome to reduce the toxicity induced by antitumour therapies and improve the response to these therapies, incorporating, for example, probiotics as an adjuvant treatment, or designing microbial enzyme target molecules [68].

### 6.1. Immunotherapy

Recently, immunotherapy has emerged as an effective therapy with favourable results in killing tumour cells [69]. For some patients with recurrent or persistent metastatic gynaecological cancer, programmed cell death-1/programmed cell death-ligand 1 (PD-1/PD-L1) inhibitors are a possible option to enhance clinical outcomes [70].

The identification and elimination of tumour cells depends on cellular immunity mediated by T cells, which, through receptors (TCRs), bind with the specific antigen of the major histocompatibility complex (MHC) on the surface of tumour cells. The interaction of TCRs and MHC is regulated by a series of immune checkpoints, serving to activate or inhibit T cells. Cytotoxic T-lymphocyte-associated protein 4 (CTLA-4), PD-1, and PD-L1 are co-inhibitors which stop the immune response to prevent autoimmune diseases. In the tumour microenvironment, tumour and stromal cells overexpress co-inhibitory ligands and receptors. Thus, the binding of the PD-1 receptor with its PD-L1 ligand transmits inhibitory signals, blocking the immune activity of T cells, thus allowing tumour cells to escape the immune system [71]. Monoclonal antibodies against PD-1 (nivolumab), PD-L1 (pembrolizumab), and CTLA-4 (ipilimumab), reactivate the immune response of patients against cancer [72].

Recent studies have described the role of the microbiome in regulating tumour responses to immunotherapies targeting PD-L1 or cytotoxic T-lymphocyte-associated protein 4 (CTLA-4). Sivan et al. conducted studies on two groups of mice with subcutaneous melanomas and different intestinal microbiota, since the two groups were bred and raised in different laboratories. They obtained evidence that one of the two groups of mice included in the trial developed immune responses through induction and infiltration of antitumour CD8+ T cells, while the other group did not. After analysing the faecal microbiota of both groups, differences in composition were observed, where a greater amount of *Bifidobacterium* was found in the group that developed an antitumour immune response. This bacterium is capable of mediating dendritic cell reactivation by itself, which promotes the CD8+ T cells’ response to eliminate tumour cells. The transfer of faecal *Bifidobacterium* in combination with the use of anti PD-L1 antibodies greatly improved the immune response, stimulating greater T cell production and helping to control the tumour [73].

In 2018, Routy et al., in a study comparing the faecal microbiota of a group of mice with non-small-cell lung cancer and renal carcinoma which responded positively to blocking immune checkpoints, and another group that did not respond, observed that in the group of mice that best responded to treatment with anti-PD-1, there was an enrichment in the groups of Firmicutes (*Clostridiales*), in addition to a significant increase in *Alistipes*, *Ruminococcus*, and *Eubacterium* species, and especially *Akkermansia muciniphila* species. In this last study, and unlike Sivan et al., Routy et al. found that the enrichment in the aforementioned species was accompanied by a relative decrease in other species, including *Bifidobacterium adolescentis*, *Bifidobacterium longum*, and *Parabacteroides distasonis*. Additionally, it was observed that the presence of *Enterococcus hirae* together with *Akkermansia muciniphila* enhanced the anti-PD-1 antitumour response in mice that best respond to anti-PD-1 antibodies. These two bacteria induce the production of IL-12, a Th1-type cytokine, in dendritic cells, stimulating the production of intestinal CD4+ T cells that express CCR9 receptors for chymosins in tumour beds, lymph nodes that drain the tumour, and in the mesenteric lymph nodes, exerting an adjuvant effect on the anti-PD-1 response. Conversely, it was observed that the group of mice that did not respond to anti-PD-1 had more *Corynebacterium aurimucosum* and *Staphylococcus haemolyticus* [74].

Gopalakrishnan et al. also investigated how gut microbiota can modulate the response to anti-PD-1. They saw that patients with metastatic melanoma, whose intestinal microbiome presented greater diversity and abundance in the Ruminococcaceae and *Faecalibacterium* families, developed a better systemic and antitumour immune response, showing greater antigen presentation, and an increase in the function of effector T cells in the periphery and the tumour microenvironment. Whereas the group of patients with metastatic melanoma, but whose intestinal microbiomes presented little diversity and a greater relative abundance in *Bacteroidales*, showed some alterations in the systemic and antitumour immune response due to limited intratumoural and myeloid lymphoid infiltration and low antigen presentation [75].

In summary, an abundance of *Clostridiales* in gut microbiome correlated to patients who respond positively to PD-1 blockade therapy, while the nonresponders’ microbiomes were enriched with *Bacteroidale* [76]. In addition, the previously mentioned studies have demonstrated that there are three species (*Bifidobacterium*, *Akkermansia muciniphila*, and *Faecalibacterium*) that could be considered to be immune adjuvants in PD-1/PD-L1 immunotherapy. However, these species do not act by themselves, as they also influence the ecology and metabolism of the intestinal microbiota in response to immunotherapy. Furthermore, to support this hypothesis, the efficacy of inhibiting immune checkpoints has been shown to be reduced in patients who have received antibiotic treatment before or after immunotherapy [77].

In 2017, Chaput et al., analysing the gut microbiome of patients affected by metastatic melanoma treated with ipilimumab, revealed that enrichment with *Bacteroidetes* had a protective role against colitis, but also a poor tumour response, while enrichment with *Faecalibacterium* genus and other Firmicutes enhanced progression-free survival [78]. In another study, Cramer et al. reported that *Bacteroides fragilis* can be considered an immunogenic bacterium that acts as an anticancer probiotic, as its polysaccharide capsule induces IL-12-dependent TH1 immune responses. This bacterium enhances the effect of immunological treatment with anti-CTLA-4 [79]. In another study, Vétizou et al. also concluded that the composition of the microbiota, specifically the abundance of *Bacteroides fragilis* and/or *Bacteroides thetaiotaomicron* and *Burkholderiales*, modulates the response to a CTLA-4 blockade. The distribution of *Bacteroides fragilis* in the intestinal mucosa and its association with *Burkholderiales* stimulates pyrine-caspase1 inflammasome formation and activates the TLR2/TLR4 signalling pathway, which could explain the immunomodulatory effects that these bacteria have on CTLA-4. Although Ipilimumab, a monoclonal antibody to CTLA-4, is highly effective in immunotherapy, it can sometimes cause colitis. To counteract this, it has been observed that oral administration of *Bacteroides fragilis* and *Burkholderia cepacia* in mice can restore the response to anti-CTLA-4 and significantly reduce colitis. The efficacy of Ipilimumab is highly dependent on intestinal microbiota, so that enrichment with *Bacteroides fragilis* is necessary for the activation of CD4+ cells and obtaining favourable treatment results [80] (Table 3).

However, whether endometrial microbiota can actually influence the efficacy of immunotherapy in endometrial cancers still needs to be investigated.

### 6.2. Chemotherapy

In patients with advanced cancers, cytotoxic drugs are used as the main therapy. However, these drugs often have strong adverse effects. Gut microbiota is considered a key element in enhancing the efficacy and reducing the toxicity of chemotherapy drugs, as well as improving the sensitivity to chemotherapy. Immunomodulation is one of the key mechanisms by which the microbiota intervenes in the response to different types of treatments.

The efficacy of cyclophosphamide, a cytotoxic alkylating agent used in chemotherapy, is modulated by the presence of Gram-positive bacteria such as *Enterococcus hirae*, *Lactobacillus johnsonii*, *Lactobacillus murinus*, and segmented filamentous bacteria. Furthermore, translocation of *Enterococcus hirae* improves the intratumoural CD8/Treg ratio. At the same time, *Barnesiella intestinihominis*, a Gram-negative bacterium, has been shown to enhance the infiltration of interferon-c-producing T cells into tumour tissue to enhance the effect of cyclophosphamide [77].

Gemcitabine is a chemotherapeutic agent belonging to the group of nucleoside (cytidine) analogues approved by the FDA to be used as a treatment for various solid tumours, including advanced endometrial cancer. This drug has the ability to kill cells in the S phase of DNA synthesis and blocks the progression of cells through the G1/S phase. Gemcitabine is metabolized into gemcitabine diphosphate and triphosphate, which, once incorporated into DNA, inhibits polymerase activity. Furthermore, apoptosis is induced through the recognition of incorporated gemcitabine by p53 and DNA-dependent kinases [81]. Chen et al. conducted a study on transgenic mice with pancreatic cancer treated with gemcitabine supplemented with *Lactobacillus*. In that study, *Lactobacillus paracasei*, a Gram-positive facultative heterofermentative lactic acid bacterium which is part of human and animal intestinal microbiota was used. *Lactobacillus paracasei* has been shown to inhibit Th2 cytokine production and modulate the Th1/Th2 balance by increasing IFN-γ levels. It was subsequently observed that the Th2 response produces tumourigenesis-promoting effects in patients with pancreatic cancer. Chen et al. used another probiotic, *Lactobacillus reuteri*, due to its antioxidant activity and ability to reduce levels of IL6 interleukin with tumorigenic activity. Gemcitabine treatment is known to cause increased levels of liver enzymes. After combining the treatment with probiotics, a decrease in the level of these enzymes was observed [82].

Irinotecan, another chemotherapeutic agent used for endometrial cancer, acts as a Topoisomerase I inhibitor [83]. Although its effectiveness as an anti-tumour treatment in various cancers is quite significant, this agent has several side effects at the gastrointestinal level, causing mucositis and diarrhoea on several occasions. Irinotecan is activated in-vivo on SN38, a potent inhibitor of Topoisomerase I, which delays the growth and proliferation of tumour and intestinal cells. SN38 is marked by glucuronic acid binding to form SN38-G, for subsequent elimination from the gastrointestinal tract. β-glucuronidase enzyme, produced by some intestinal bacteria, is capable of eliminating glucuronic acid from SN38-G, reactivating it to SN38, thus causing epithelial damage, shedding, diarrhoea, and weight loss in animal models. Bhatt et al., based on the relationship between SN38-G activation, β-glucuronidase activity, and intestinal toxicity produced by Irinotecan treatment, decided to investigate the effect of inhibiting this enzyme’s activity to alleviate toxicity. In that study, it was observed that the use of amoxapine and pyrazolo 4-3-c quinoline, inhibitors of β-glucuronidase activity, protects the gastrointestinal epithelium by reducing the production of pro-inflammatory cytokines, and improves the response to treatment with Irinotecan. In addition, Irinotecan induces changes in the composition of the intestinal microbiota, increasing, above all, the levels of *Proteobacteria* (*Enterobacteriaceae*), in addition to *Verrucomicrobia* and *Akkermansia muciniphila*. The use of *GUS* gene (a gene that encodes the enzyme β-glucuronidase) inhibitors has been shown to reduce *Proteobacteria* levels [84].

Doxorubicin is a chemotherapeutic anticancer drug belonging to the anthracycline family, used to treat various types of cancers, including endometrial cancer. It is characterized by its ability to inhibit the growth of both cancer cells and bacteria through the generation of free radicals, DNA intercalation, alkylation and cross-linking of proteins, interference with DNA unwinding and Topoisimerase II, and direct membrane damage. However, the use of drugs belonging to the anthracycline family leads to the accumulation of toxic metabolites in healthy tissue [85]. In addition to the heart, the gut is also affected by the toxicity associated with the use of Doxorubicin. This drug causes damage to the intestinal epithelium by inducing apoptosis in the epithelial cells of the jejunum and damage to the mucosa, reducing the proliferation of crypts, so that fewer crypts are formed, and with smaller villi [86]. Oral mucositis, another reaction associated with doxorubicin-induced toxicity, produces an increase in salivary flow, gum inflammation, and sore formation. Oral mucositis produces dysbiosis, decreasing the levels of the *Streptococcus*, *Actinomyces*, *Gemella*, *Granulicatella*, and *Veillonella* genera, and increasing the levels of other Gram-negative bacteria such as *Fusobacterium nucleatum* and *Prevotella oris*. *Fusobacterium nucleatum* have pro-inflammatory and pro-apoptotic activity, contributing to the damage produced in the mucosa [87]. Conversely, bacteria of the intestinal microbiota have been implicated in the inactivation of some drugs, including doxorubicin. Yan et al. identified *Raoultella planticola* as a powerful inactivator of doxorubicin under anaerobic conditions, and demonstrated that this bacterium deglycosylates doxorubicin into the metabolites 7-deoxydoxorubicinol and 7-deoxydoxorubicinolone by the reductive deglycosylation mechanism. Subsequently, doxorubicin was anaerobically degraded by *Klebsiella pneumoniae* and *Escherichia coli* [85].

Paclitaxel, another chemotherapeutic agent used in the treatment of endometrial cancer, has neurological side effects, producing peripheral neuropathies. Ramakrishna et al. proposed that Paclitaxel lowers beneficial bacteria levels such as *Akkermansia muciniphila* which promotes barrier function. In addition, they observed that the *Porphyromonadaceae* family is involved in the dysbiosis produced by Paclitaxel, which, in turn, has been implicated in neurological damage produced in glial cells [88] (Table 3).

### 6.3. Radiotherapy

Radiation therapy is an effective method of antitumour treatment, based on the genotoxic effect on tumour cells, and through which cell death is induced by local irradiation, accompanied by systemic immunity and inflammation [89]. However, irradiation-mediated intestinal toxicity was observed in several cases, which involves an alteration in the composition of the microbiota, and leads to dysfunction of the intestinal barrier and apoptosis in intestinal crypts [90]. Yan et al. identified two soluble proteins produced by *Lactobacillus rhamnosus*, p75 and p40, which induce the activation of the AKT signalling pathway by stimulating cell proliferation, and inhibit the apoptosis induced by tumour necrosis factor, in epithelial cells. In addition to *Lactobacillus rhamnosus*, *Lactobacillus casei* and *Lactobacillus acidophilus* have also been shown to have protective roles in minimizing the damage caused by radiation therapy [91]. Ciorba et al. also found that administration of *Lactobacillus rhamnosus* before radiotherapy decreases epithelial apoptosis and stimulates crypt survival in mice guts. The cell wall of this bacterium, like all Gram-positive bacteria, is composed of peptidoglycan and lipoteichoic acids which act as Toll-like receptor-2 (TLR-2) ligands. The activation of TLR-2 leads to COX-2 expression and reactive oxygen species (ROS) production to activate the cytoprotective system NRF-2, a transcription factor that regulates the expression of detoxifying and antioxidant enzymes, thereby contributing to the protection of intestinal cells from damage caused by radiotherapy [92].

While Ciorba et al. did not find any radioprotective effect of *Bifidobacterium* [92], Delia et al., in a previous study, proved that VSL3, a mixture comprised of *Lactobacillus* (*Lactobacillus casei*, *Lactobacillus plantarum*, *Lactobacillus acidophilus*, and *Lactobacillus delbrueckii* subsp. *bulgaricus*), *Bifidobacteria* (*Bifidobacterium longum*, *Bifidobacterium breve*, and *Bifidobacterium infantis*), and a species of *Streptococcus salivarius* subsp. *Thermophilus*, reduced the toxicity (diarrhoea) produced by radiotherapy [93] (Table 3).

### 6.4. Targeted Therapy

Recently, in addition to the previously mentioned traditional therapies, targeted molecular therapies have proven to be of essential importance in improving the long-term survival of cancer patients with specific biomarkers [94]. Trastuzumab, an FDA-approved drug which contains monoclonal antibodies targeting the *HER-2* receptor extracellular domain, is being tested for endometrial cancer, because in serous endometrial carcinoma the *HER-2* gene, which is responsible for the increase of cell proliferation, differentiation, and migration, is overexpressed [11]. Recently, Di Modica et al. analysed the gut microbiota composition of a group of breast cancer patients who responded favourably to adjuvant treatment with Trastuzumab. They found that in those patients with favourable results in response to treatment, *Clostridiales* (*Lachnospiraceae*), *Bifidobacteriaceae*, *Turicibacteraceae,* and *Bacteroidales* (*Prevotellaceae*) predominated, while in the other group of patients who did not respond to treatment, an enrichment in the phylum *Bacteroidetes* (*Bacteroidia*) was observed [95].

Erlotinib and gefitinib, two epidermal growth factor receptor (EGFR) and tyrosine kinase inhibitors, have also been tested in patients with endometrial cancer, as *EGFR* is overexpressed in 40–46% of Type I endometrial carcinoma cases, and in 34% of Type II endometrial carcinomas [11]. Flórez el al. demonstrated that 34 species of lactic acid bacteria, *Bifidobacteria,* and other intestinal bacteria are resistant to treatment with erlotinib and gefitinib, so that the abundance of these species was not altered after treatment [96].

Letrozole, an aromatase inhibitor that inhibits the production of local and circulating estrogens, has also been used in clinical trials as a treatment for endometrial cancer [97]. Cao et al., in a study with mice, found that Letrozole treatment produces a decrease in the Firmicutes/Bacteroidetes ratio, contributing to a decrease in inflammation. Furthermore, in the same study, it was observed that Letrozole altered the diversity of the intestinal microbiota in mice, significantly decreasing Ruminococcaceae levels [98] (Table 3).

### 6.5. Toxicity

It is evident that the intestinal microbiota can modulate the response to different antitumour treatments. However, in turn, the microbiota is itself altered in response to treatment [99]. Treatment of endometrial cancer can cause several symptoms in patients, one of which is vaginal atrophy, caused by cell damage as a result of radiation therapy. Patients with vaginal atrophy have less *Lactobacillus*, the first line of defense in the female urogenital tract. Damage to the vaginal epithelium caused by radiation therapy allows pathogens to penetrate the epithelium and causes inflammation that ultimately contributes to vaginal atrophy [100].

Chemotherapy, in turn, causes several side effects in patients, including gastrointestinal mucositis, which results in several symptoms in patients, such as nausea, diarrhoea, vomiting, and abdominal pain. Gastrointestinal mucositis is a lesion characterized by atrophy of the villi and the loss of enterocytes, which leads to epithelium deterioration and gut-barrier alteration [101]. Gut microbiota has been implicated in many of the pathological aspects of gastrointestinal mucositis caused by chemotherapy. After chemotherapy, the permeability of the intestinal mucosa increases due to the atrophy of the villi as a consequence of gastrointestinal mucositis. However, intestinal microbiota, especially *Bifidobacteria* and *Lactobacillus*, improves the functioning of the epithelial barrier, reducing its permeability by binding to TLR-2 receptors, which leads to protein kinase C phosphorylation and the production of proteins that form tight junctions. The levels of these bacteria and others involved in maintaining the normal permeability of the epithelial barrier (*Faecalibacterium*, *Ruminococcus*, *Coprococcus*, *Dorea*, *Lachnospira*, *Roseburia*, *Clostridium* and *Bifidobacterium*) decrease after chemotherapy, which explains the increased permeability of the intestinal mucosa as a result of mucositis [101,102]. Additionally, in another study, Montassier et al. noticed a decrease in both the number and diversity of intestinal microbiota after chemotherapy, which is associated with an increase in *Bacteroides*, *Enterococci,* and Enterobacteriaceae, and a decrease in Firmicutes (Ruminococcaceae, Lachnospiraceae) and Actinobacteria (*Bifidobacterium*). In the same study, they showed that bacteria that modulate the NFҡB signalling pathway to decrease the inflammatory response, such as *Faecalibacterium*, *Ruminococcus*, *Coprococcus*, *Dorea*, *Lachnospira*, *Roseburia* and *Clostridium*, decreased after chemotherapy, as did *Bifidobacterium*, whose function under normal conditions is to inhibit the inflammatory response. The decrease in these bacteria, which are also butyrate producers, implies a decrease in the production of short-chain fatty acids, and consequently the inflammatory response is not inhibited. Intestinal mucosa composition is also altered after chemotherapy. As a result of the reduction of butyrate-producing bacteria, butyrate is not produced, and therefore, mucin synthesis via MUC2 is not stimulated, which leads to tissue damage and translocation of bacteria due to alterations in the composition of the intestinal mucosa.

Conversely, an increase in *Citrobacter* was observed after chemotherapy. This bacterium stimulates NFҡB production and therefore stimulates the inflammatory response. In addition, it also participates in intestinal barrier degradation, using mucinases and glucosidases to digest mucin [101] (Table 3).

## 7. Modulation of Endometrial Microbiota

Commensal bacteria can protect their host from pathogen infections due to their ability to better adapt to the environment than pathogens, which allows them to compete successfully. In addition, they make better use of the available nutrients, leaving the pathogens without an energy source [103]. Consequently, there is a growing interest in modulating the endometrial microbiome composition and environment to break dysbiosis and prevent endometrial diseases.

Probiotics are live microorganisms that confer health benefits to their host. These bacteria can produce bioactive molecules that act on the body, promoting good health, with low toxicity and few side effects. Most of the studies carried out selected the *Lactobacillus rhamnosus* BPL005 strain as the best candidate to improve the female reproductive tract, due to its capacity in-vitro to reduce pH levels and produce organic acids such as lactate, which promotes the reduction of pathogenic bacteria [104]. Chenoll et al., with the aim of investigating whether strain *Lactobacillus rhamnosus* BPL005 could have beneficial effects against endometrial infections caused by pathogens, used human endometrial epithelial cells (HEEC) co-cultured with pathogenic bacteria (*Atopobium vaginae*, *Gardnerella vaginalis*, *Propionibacterium acnes*, and *Streptococcus agalactiae*) alone, and in combination with the strain *Lactobacillus rhamnosus* BPL005. The study showed that in the HEEC cells cultured with the strain *Lactobacillus rhamnosus* BPL005, there was a reduction of the pH, being less than 5. This low pH limits the growth of pathogens and inhibits their adhesion to endometrial cells. Another finding confirmed that *Lactobacillus rhamnosus* BPL005 decreased the levels of some metabolites like propionic acid produced by *Propionibacterium acnes* (linked to symptomatic bacterial vaginosis profiles) in endometrial cell cultures, leading to a drift towards a healthy organic acid profile. Furthermore, lactic acid produced by *Lactobacillus rhamnosus* BPL005 had a bactericidal effect against pathogen colonization in HEEC cells. These effects on pH and organic acid production were considered to be pathogen inhibition pathways to decrease pathogen colonization. Additionally, the *Lactobacillus rhamnosus* BPL005 strain produced bacteriocins, further protecting against vaginal pathogens [105].

Female genital microbiota modulation could also be used to protect against infection. Bacterial vaginosis is linked to endometrial microbial colonization, and a recent study found a polymicrobial *Gardnerella vaginalis* biofilm in the uterus of women with bacterial vaginosis [106]. The addition of the *Lactobacillus rhamnosus* BPL005 strain to HEEC cells colonized by pathogens increased proinflammatory cytokines such as IL-1RA and IL-1β, and decreased the proinflammatory IL-6, IL-8, and MCP-1 cytokines, which were previously increased due to the pathogens’ presence [105].

Recent studies have also shown that probiotic lactobacilli (*Lactobacillus reuteri RC-14* and *Lactobacillus rhamnosus GR-1*) can improve endometrial epithelial cells’ barrier- function in response to the human immunodeficiency virus-1 (HIV-1). These bacterial strains are able to modulate the immune profile, indicating that female reproductive tract microbiota could be an important factor in the acquisition of resistance to viruses [107].

Prebiotics are compounds that serve as nutrients and promote the growth and activity of beneficial microorganisms with the aim of enhancing health. Lactoferrin is a prebiotic agent used to modify the endometrial microbiome. Lactoferrin, orally administrated during and after antibiotics treatment in women undergoing infertility treatment, can increase *Lactobacillus* levels in the endometria of non-*Lactobacillus* dominant patients after three months of use [108]. In addition, Lactoferrin administration showed effective results against bacterial vaginosis, preventing endometrial infections [104].

The use of prebiotics and probiotics can provide greater benefits than the use of antibiotics alone, which produces short-term results but which aggravates dysbiosis and promotes resistance over the long-term.

Finally, vaginal microbiota transplants (VMTs) (the transfer of cervicovaginal fluid from a healthy donor to a patient to restore their microenvironment) to patients suffering from symptomatic and recurrent vaginosis as a therapeutic alternative has shown positive treatment outcomes [109]. Currently, two Phase I/II clinical trials in the USA (NCT03769688 and NCT04046900) and one in Israel (NCT02236429) are recruiting participants to analyse the efficacy and safety of VMTs in women with bacterial vaginosis.

VMTs could be an effective tool for managing endometrial dysbiosis, as uterine colonization by microorganisms through vaginal-cervical ascension has been described previously [110,111].

VMTs could be used to modulate the vaginal microbiome by restoring the microenvironment for the prevention of endometrial cancer. Nevertheless, future studies with larger cohorts and randomized, placebo-controlled studies will be necessary to determine the efficacy and durability of VMTs for endometrial cancer prevention.

## 8. Conclusions

Due to the importance of the microbiome in many human physiological processes and recent advances in highly sensitive molecular techniques which facilitate the identification of microorganisms, several emergent studies have shown interest in investigating the relationship between gut and endometrial microbiome in endometrial cancer, one of the most common cancers in women worldwide, which occurs more frequently after menopause. There is evidence that the presence of both *Atopobium vaginae* and *Porphyromonas somerae* in the gynaecological tract is statistically related to endometrial cancer, particularly when vaginal pH is high. In endometrial cells, these two bacteria can also induce the production of IL-1α, IL-1β, IL-17α, and TNFα, pro-inflammatory cytokines which are involved in the carcinogenesis of various tumours. Because endometrial cancer is estrogen-dependent, an excess of estrogen in the body is considered to be an important risk factor for endometrial cancer. In this context, estrobolome (an aggregate of enteric bacterial genes whose products are capable of metabolizing estrogens) plays a fundamental role. These bacteria with β-glucuronidase activity can activate conjugated estrogens, transported from the liver to the intestine, though deconjugation. Consequently, estrobolome dysbiosis can lead to an estrogen increase, contributing to carcinogenesis. The microbiome is also involved in the body’s response to treatment, so it can alleviate some of the side effects of various antitumour therapies and reduce their toxicity. However, the microbiome can also be altered in response to treatment. Due to the implication of the microbiome in various processes such as inflammation, estrogen metabolism, carcinogenesis, and antitumour treatments, we can conclude that modulating gut and endometrial microbiome in combination with traditional endometrial cancer treatments may provide an alternative method to achieve better antitumour therapy results and improve patient living conditions. Further research into metagenomic analysis in endometrial cancer is needed to improve our knowledge of this topic and to discover novel markers with therapeutic implications.

## Figures and Tables

**Figure 1 jpm-11-00659-f001:**
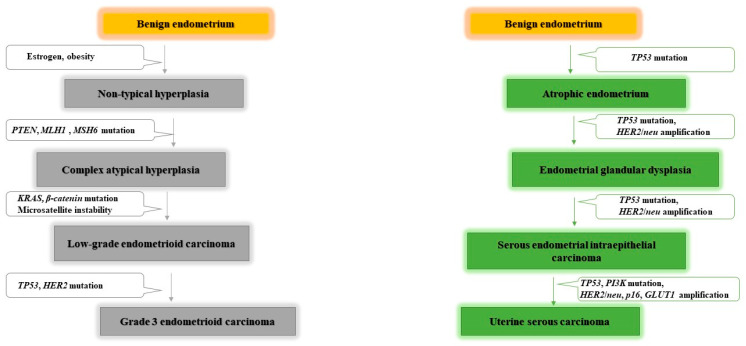
Commonly altered genes in endometrial carcinogenesis. Endometrioid carcinoma is estrogen dependent, and obesity is associated with an elevated endometrioid cancer risk and mortality. Several mutations can lead to initiation and development of endometrioid carcinoma, such as *PTEN*, *KRAS* and *β-catenin* mutations, wheras non-endometrioid carcinoma more often harbours mutations in *TP53*. *PTEN*: phosphatase and tensin homolog. *MLH1*: human mutL homolog 1, involved in DNA mismatch repair. *MSH6*: human mutS homolog 6, involvedd in DNA mismatch repair. *KRAS*: *KRAS* proto-oncogene, GTPase. *β-catenin*: CTNNB gene (cadherin associated protein) a signaling molecule involved in the control of cell growth and differentiation. TP53: tumour protein p53, tumour supressor. *HER2/neu*: Erb-B2 receptor tyrosine kinase 2, proto-oncogen. PI3K: phosphatidylinositol 3-kinase, proto-oncogen. P16: cyclin-dependent kinase inhibitor 2A gene, tumour supressor. Glut1: glucose transporter 1.

**Figure 2 jpm-11-00659-f002:**
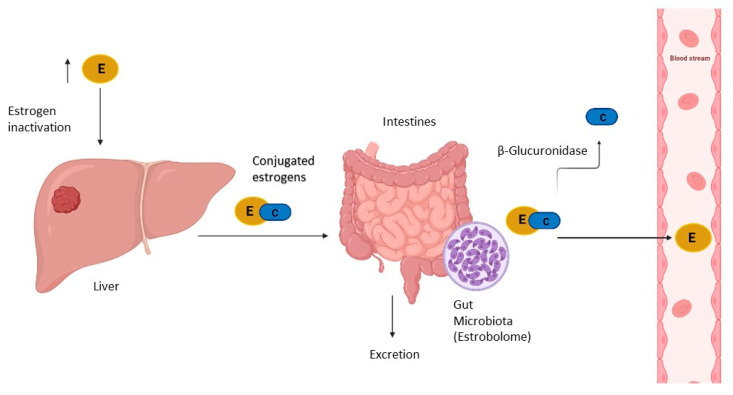
Estrogen metabolism involving gut microbiota (estrobolome). Estrogens are inactivated in liver through conjugation for further excretion. However, some of inactivated estrogens are reabsorbed into the bloodstream across activation in intestines by estrobolome. E: estrogen; C: conjugation with glucuronide acid binding.

**Figure 3 jpm-11-00659-f003:**
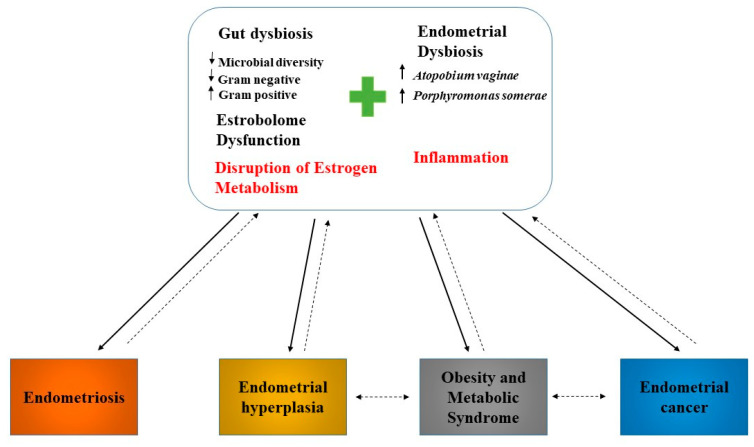
Endometrial and gut dysbiosis implication in endometrial cancer induction. The simultaneous presence of *Atopobium vaginae* and *Porphyromonas somerae* in the endometrium induce the production of pro-inflammatory cytokines, which are involved in endometrial carcinogenesis and other endometrial diseases.

**Table 1 jpm-11-00659-t001:** Endometrial microbiome characterization studies.

Author	Year	Sample Size	Sample Type	Methods	Finding
Mitchell et al.	2018	Women underwent hysteroctomy for benign disease without cancer indications (*n* = 58).	Vaginal and endometrial swabs.	Bacterial 16S rRNA sequencing.	↑*Lactobacillus iners* (45%), *Lactobacillus crispatus* (33%), *Gardnerella vaginalis*, *Prevotella* spp., *Atopobium vaginae*, and *Sneathia*.
Fang et al.	2016	-Patients with only endometrial polyps (*n* = 10). -Patients with both endometrial polyps and chronic endometritis (*n* = 10). -Healthy women (*n* = 10).	Vaginal and endometrial swabs.	Bacterial 16S rRNA genes sequencing.	↑*Lactobacillus*, *Gardnerella*, *Bifidobacterium*, *Streptococcus*, and *Alteromonas* in healthy group compared to endometrial polyps and chronic endometriosis group.
Verstraelen et al.	2016	Women with various reproductive conditions, without uterine anomalies (*n* = 90).	Endometrial biopsy: tissue and mucus.	16S rRNA gene V1–2 region	Uterine microbiome dominated by *Bacteroides* (*B. xylanisolvens*, *B. thetaiotaomicron*, and *B. fragilis*) and *Pelomonas*.
Chen et al.	2017	Reproductive age women operated for conditions not known to involve infection (*n* = 95).	Endometrial swab and tissue.	16S rRNA amplicon sequencing	↑*Pseudomonas*, *Acinetobacter*, *Vagococcus*, and *Sphingobium* in the endometrium
Winters et al.	2019	Women (*n* = 25) underwent a hysterectomy for fibroids (23) and endometrial hyperplasia (2).	Endometrial swab.	Sequencing of the 16S rRNA gene.	↑*Acinetobacter*, *Pseudomonas*, Comamonadaceae, and *Cloacibacterium* in endometrium.
Lu et al.	2020	Women undergone a hysterectomy for benign disease and any stage of endometrial cancer (*n* = 50).	Endometrial tissue.	16S rRNA gene sequencing for bacterial communities.	↑*Rhodococcus*, *Phyllobacterium*, *Sphingomonas*, *Bacteroides*, and *Bifidobacterium*.

**Table 2 jpm-11-00659-t002:** Studies of intratumoural microbiota in endometrial cancer patients.

Author	Year	Simple Size	Sample Type	Methods	Finding
Walther-António et al.	2016	-Patients with endometrial cancer (*n* = 17). -Patients with endometrial hyperplasia (*n* = 4). -Patients with benign uterine conditions (*n* = 10).	Endometrial swab and scrabe.	16S rRNA sequencing of V3-V5 region.	Patients with endometrial cancer: ↑Firmicutes (*Anaerostipes*, *ph2*, *Dialister*, *Peptoniphilus*, *1–68*, *Ruminococcus*, and *Anaerotruncus*), Spirochaetes (*Treponema*), Actinobacteria (*Atopobium*), Bacteroidetes (*Bacteroides* and *Porphyromonas*), and Proteobacteria (*Arthrospira*). Close correlation between *Atopobium vaginae* and *Porphyromonas* spp. and endometrial cancer, especially when the vaginal pH is high (˃4.5).
Lu et al.	2020	Patients undergone a hysterectomy for benign disease and any stage of endometrial cancer (*n* = 50).	Endometrial cancer tissue.	16S rRNA gene sequencing for bacterial communities.	↓Local microbiome diversity in patients with endomatrial cancer. ↑*Micrococcus* asociated with an inflammatory profile in endometrial cancer patients.
Gonzalez-Bosquet et al.	2021	-Patients with high grade serous ovarian cancer (HGSC) (*n* = 112). -Patients with endometrioid endometrial cancer (EEC) (*n* = 62). -Women with normal fallopian tubes, and no risk factors for cancer (*n* = 12).	Frozen ovarian and endometrial tumour tissue.	16S rRNA gene sequencing.	-93 bacterial, archaea, and viral transcripts (BAVTs) were differentially expressed between HGSC and EEC. -The diversity of BAVT species decreased in EEC, and even more in HGSC compared to normal samples.
Walsh et al.	2019	Patients with a variety of uterine conditions (*n* = 148): -Patients without endometrial cancer (*n* = 75). -Patients with Type I endometrial cancer (*n* = 56). -Patients with Type II endometrial cancer (*n* = 10). -Patients with complex atypical hyperplasia (*n* = 7).	Uterine, fallopian and ovarian samples (swabs and scrapes)	-Amplification and sequencing of V3-V5 region of the 16S rRNA gene.	In endometrial cancer patients: ↑*Porphyromonas somerae.* *↑**Anaerococcus tetradius*, *Anaerococcus lactolyticus*, *Peptoniphilus coxii*, and *Campylobacter ureolyticus* related to postmenopause status, facilitating the subsequent colonization by *Porphyromonas somerae*.

**Table 3 jpm-11-00659-t003:** Gut microbiota modulation of antitumoural therapies efficacy and toxicity.

Antitumoural Therapy		Author	Year	Sample Type Analized	Methods	Finding
Immunotherapy	anti-PD-L1	Sivan et al.	2015	Two groups of mice with subcutaneous melanomas and different intestinal microbiota, from different laboratories.	Transfer of faecal material within both group of mice before tumour implantation. 16S rRNA sequencing analysis	Gut microbiome of responder group: ↑*Bifidobacterium*
	anti-PD-1	Routy et al.	2018	Mice with non-small-cell lung cancer and renal carcinoma (responders to blocking immune checkpoints, and non-responders).	Faecal microbiota transplantation and 16S ribosomal rRNA of faecal samples	Gut microbiome of responder group: ↑Firmicutes (*Clostridiales*), ↑*Alistipes*, ↑*Ruminococcus*, ↑*Eubacterium* spp. ↑*Akkermansia muciniphila* ↑*Enterococcus hirae*, ↓*Bifidobacterium adolescentis*, ↓*Bifidobacterium longum* ↓*Parabacteroides distasonis* Gut microbiome of non-responder group: ↑*Corynebacterium aurimucosum*, ↑*Staphylococcus haemolyticus*
		Gopalakrishnan et al.	2018	Patients with metastatic melanoma (*n* = 112)	16S rRNA gene sequencing of oral, buccal and faecal samples, and tumour biopsies at treatment initiation and 6 months after treatment initiation.	Gut microbiome of responders group: ↑Diversity, ↑*Ruminococcaceae* ↑*Faecalibacterium* Non-responders group gut microbiome: ↓Diversity, ↑Bacteroidales
	anti-CTLA-4	Chaput et al.	2017	Patients with metastatic melanoma (*n* = 26)	16S rRNA gene sequencing at baseline and before each ipilimumab infusion in faecal samples.	Gut microbiome of responders group: ↑*Faecalibacterium*, ↑other Firmicutes Non-responders gut microbiome: ↑Bacteroidetes
		Cramer et al.	2017	Patients with metastatic melanoma and received ipilimumab (*n* = 34)	16S rRNA gene amplification and multiparallel sequencing of faecal samples.	*Bacteroides fragilis* enhances the effect of immunological treatment with anti-CTLA-4
		Vétizou et al.	2015	Mice with sarcomas	High-throughput pyrosequencing of 16S rRNA gene amplicons of faeces.	Gut microbiome of responders group: ↑*Bacteroides fragilis* ↑*Bacteroides* thetaiotaomicron ↑*Burkholderiales* ↑*Burkholderia cepacia*
Chemotherapy	cyclophosphamide	Ma et al.	2019	Mice treated with cyclophosphamide	High-throughput 454 pyrosequencing in faecal samples. Quantitative PCR targeting the domain bacteria and specific bacterial groups.	Gut microbiome involved in enhacing treatment response: *Enterococcus hirae* *Lactobacillus johnsonii* *Lactobacillus murinus* Segmented filamentous bacteria *Barnesiella intestinihominis*
	Gemcitabine	Chen et al.	2020	Transgenic mice with pancreatic cancer	Probiotic oral gavage of *Lactobacillus paracasei* and *Lactobacillus reuteri*. 16S rRNA Amplicon Sequencing.	↑*Lactobacillus paracasei*, *Lactobacillus reuteri* enhacing treatment response in transgenic mice.
	Irinotecan	Bhatt et al.	2020	Tumour xenograft model	16S rRNA Amplicon Sequencing	Changes in the composition of the intestinal microbiota induced by irinotecan: ↑Proteobacteria (*Enterobacteriaceae*), ↑Verrucomicrobia ↑*Akkermansia muciniphila*
	Doxorubicin	Hong et al.	2019	Patients with chemotherapeutic treatment for a solid tumour (*n* = 30) and non-cancer controls (*n* = 30)	Amplification and sequencing of 16S rRNA gene and ITS 1 DNA	Gut dysbiosis induced by doxorubicin: ↓*Streptococcus*, ↓*Actinomyces* ↓*Gemella*, ↓*Granulicatella* ↓*Veillonella*, ↑*Fusobacterium nucleatum* ↑*Prevotella oris*
		Yan et al.	2018	Healthy donors	16S rRNA Amplicon Sequencing of faecal samples	↑*Raoultella planticola*, *Klebsiella pneumoniae* and *Escherichia coli* involved in doxorubicin inactivation and degradation.
	Paclitaxel	Ramakrishna et al.	2019	Two groups of mice: sensitive and resistant to Paclitaxel-induced pain.	16S rRNA gene sequencing.	↓*Akkermansia muciniphila*
	Toxicity induced by chemotherapy	Van Vliet et al.	2010	Mice colonic tissues	Elisa for measurement of intestinal permeability. PCR Mucosal cytokine measurements	Gut dysbosis associated to toxicity induced by chemotherapy: ↓*Bifidobacteria* ↓*Lactobacillus*, ↓*Faecalibacterium* ↓*Clostridium*
		Montassier et al.	2015	Patients with non-Hodgkin’s lymphoma who received the same myeloablative conditioning regimen and no other concomitant therapy such as antibiotic (*n* = 28)	Amplification and sequencing of 16S rRNA genes in faecal samples	Gut dysbosis associated to toxicity induced by chemotherapy: ↑*Bacteroides*, ↑*Enterococcus* ↑Enterobacteriaceae ↓Firmicutes (*Ruminococcaceae*, Lachnospiraceae), ↓Actinobacteria (*Bifidobacterium*), ↑*Citrobacter*, ↓*Ruminococcus*, ↓*Coprococcus*, ↓*Dorea*, ↓*Lachnospira*, ↓*Roseburia*
Radiotherapy		Yan et al.	2007	Colon organ culture	Purification and analizing of proteins from *Lactobacillus*	*Lactobacillus rhamnosus*, *Lactobacillus casei* and *Lactobacillus acidophilus* have protective roles in minimizing the damage caused by radiation therapy
		Ciorba et al.	2012	Mice small intestine	Protein and nucleic acid analysis	Administration of *Lactobacillus rhamnosus* before radiotherapy decreases epithelial apoptosis and stimulates crypt survival in mice guts
		Delia et al.	2007	Patients who underwent surgery for sigmoid, rectal, or cervical cancer (*n* = 429)	Probiotic oral gavage of *Lactobacillus* spp. and *Bifidobacterium* spp.	Microbiome that can reduce gut toxicity induced by radiotherapy: *Lactobacillus* (*L. casei*, *L. plantarum*, *L. acidophilus*, and *L. delbrueckii subsp. bulgaricus*) Bifidobacteria (*B. longum*, *B. breve*, and *B. infantis*)
Targeted therapy	Trastuzumab	Di Modica et al.	2021	Female mice with breast cancer Patients with breast cancer (*n* = 24)	Faecal microbial transplantation in mice and faecal sample analysis of variable region V3 and V4 of the 16S rRNA gene for mice and human.	Gut microbiome of responders: ↑Clostridiales (Lachnospiraceae), ↑Bifidobacteriaceae, ↑Turicibacteracea, ↑Bacteroidales (Prevotellaceae) Gut microbiome of non-responders: ↑Bacteroidetes (Bacteroidia)
	Erlotinib and gefitinib	Flórez et al.	2016	Bacterial strains	Determination of minimum inhibitory concentrations	34 species of lactic acid bacteria, Bifidobacteria, and other intestinal bacteria are resistant to treatment with erlotinib and gefitinib
	Letrozole	Cao et al.	2020	Rats (8 normal controls and 30 with endometriosis)	Amplification and sequencing of 16S rRNA genes in faecal samples	↓Firmicutes/Bacteroidetes ratio, ↓inflammation, ↓Ruminococcaceae

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
