# Peer review of "Gut and Endometrial Microbiome Dysbiosis: A New Emergent Risk Factor for Endometrial Cancer"

_jpm, 2021, doi:10.3390/jpm11070659_

Round 1

Reviewer 1 Report

Correct proofreading in lines: 27, 48, 58,128, 196, 202, 272, 274, 334. 

Gene names and Latin names of microorganisms must be written in italics, for example lines: 88, 89, 280, 377, 595, 604, 701.

Abbreviations need to be expanded.

There is no reference to images in the text. Figure 2 needs to be adjusted: reduce the blood vessel and enlarge the liver and intestinal areas. There is no need to repeat the description of the picture if everything is already given in the text. Also in figure 1 need to correct gene names.

In the introduction should reveal which genes and their changes lead to Lynch syndrome.

Data should be more systematised. Chapters 2 and 3 text should provide data in table form: author, year, sample size, bacterial species, methods, etc. In chapter 6 also needed table: author, year, sample size, treatment, bacterial species, methods, etc. The information will appear more concise and easier for the reader to understand. Also, the information in the table should not be repeated in the text.

Author Response

First of all, we would like to thank the reviewer for his/her useful comments and suggestions, which undoubtedly have helped to improve our manuscript.

Comment: Correct proofreading in lines: 27, 48, 58,128, 196, 202, 272, 274, 334. 

Response: We thank the reviewer for his/her comment .The proofreading in lines 27, 48, 58,128, 196, 202, 272, 274, 334  have been corrected in the revised manuscript.

Comment : Gene names and Latin names of microorganisms must be written in italics, for example lines: 88, 89, 280, 377, 595, 604, 701.

Response: Thank you for your remark. Gene names and Latin names of microorganisms (genera and species) have been revised throughout the manuscript and they have been italicized.

Comment: Abbreviations need to be expanded.

Response: Abbreviations have been expanded throughout the revised manuscript

Comment: There is no reference to images in the text. Figure 2 needs to be adjusted: reduce the blood vessel and enlarge the liver and intestinal areas. There is no need to repeat the description of the picture if everything is already given in the text. Also in figure 1 need to correct gene names.

Response: As indicated by the reviewer, we have reference the images in the text. The figure 2 have adjusted following the reviewer's recommendations. Finally, the descriptions of the figures have been removed to avoid repetition with the text. And already we have corrected gene names in figure 1.

Comment : In the introduction should reveal which genes and their changes lead to Lynch syndrome

Response : Thank you for your comment. We have added a paragraph in the introduction section (Lines 56-64) with the genes responsible for the development of the Lynch syndrome.

« This syndrome is caused by a loss-of-function germline mutation in one of four genes (Human mutL homolog 1 (MLH1), MSH2, MSH6, and PMS1 Homolog 2 (PMS2)) involved in mismatch-pair recognition and initiation of repair [9]. MLH1 and MSH2 mutations are more frequent (60-80%) in patients with lynch syndrome comparated to MSH6 and PMS2 mutations. Mutation in epithelial cellular adhesion molecule (EPCAM) (gene located in MSH2 gene promoter and that lead to its epigenetic inactivation) is also identificated in lynch syndrome. The mismatch repair genes inactivation induces accumulation of different gene mutations, leading to cancer development with microsatellite instability phenotype [10].  » 

Comment: Data should be more systematised. Chapters 2 and 3 text should provide data in table form: author, year, sample size, bacterial species, methods, etc. In chapter 6 also needed table: author, year, sample size, treatment, bacterial species, methods, etc. The information will appear more concise and easier for the reader to understand. Also, the information in the table should not be repeated in the text.

Response: As suggested, in this new version of the manuscript we have included a table for a more concise and easier reading of the data in the chapters 2 (Table 1), 3 (Table 2)  and 6 (Table 3).

Reviewer 2 Report

The Authors described each and every section of the submitted manuscript very well. I have some minor comments regarding the submitted manuscript to improve the quality and visibility of the manuscript:

  1. Please merge the paragraph from line 48-53 and 54-58.
  2. please mention all the related Gut and endometrial microbiome with references in a tabulated form, which will be a good quick view to understand which microbiome are involved in the specific organs.
  3. please mention all different kind of microbiota involved in gynaecological conditions in a tabulated form with references, which will be helpful to understand better about the role of microbiota involved in all the related conditions.

Author Response

Comment: The Authors described each and every section of the submitted manuscript very well. I have some minor comments regarding the submitted manuscript to improve the quality and visibility of the manuscript:

Response: First of all, we thank the reviewer for his/her comments and for offering us a constructive review of our manuscript.

Comment: Please merge the paragraph from line 48-53 and 54-58.

Response : Thank you for your remark. We have merged the paragraph from line 48-53 and 54-58.  Now lines (49-56) in the revised manuscript.

Comment: Please mention all the related Gut and endometrial microbiome with references in a tabulated form, which will be a good quick view to understand which microbiome are involved in the specific organs.

Response: As indicated by the reviewer, we have now added in the revised manuscript  two tables (Table 1 and 3) for the studies about gut and endometrial microbiome (chapter 2 and 6), which allow the readers a better understant of which microbiome are involved in the specific organs.

Comment: Please mention all different kind of microbiota involved in gynaecological conditions in a tabulated form with references, which will be helpful to understand better about the role of microbiota involved in all the related conditions.

Response: As recommended by the reviewer, we have added in the revised manuscript a Table 2  of the chapter 3, that allow the readers a good understand of  the role of microbiota involved in all the related conditions.